# Congenital Malformations in Sea Turtles: Puzzling Interplay between Genes and Environment

**DOI:** 10.3390/ani11020444

**Published:** 2021-02-08

**Authors:** Rodolfo Martín-del-Campo, María Fernanda Calderón-Campuzano, Isaías Rojas-Lleonart, Raquel Briseño-Dueñas, Alejandra García-Gasca

**Affiliations:** 1Department of Oral Health Sciences, Faculty of Dentistry, Life Sciences Institute, University of British Columbia, Vancouver, BC V6T 1Z3, Canada; rodolfo@dentistry.ubc.ca; 2Marine Turtle Programme, Instituto de Ciencias del Mar y Limnología-UNAM-FONATUR, Mazatlán, Sinaloa 82040, Mexico; calderon.mfernanda@gmail.com (M.F.C.-C.); raquel@ola.icmyl.unam.mx (R.B.-D.); 3Universidad Central “Martha Abreu” de las Villas (IRL), CUM Remedios, Villa Clara 52700, Cuba; irlleonart@nauta.cu; 4Banco de Información sobre Tortugas Marinas (BITMAR), Unidad Académica Mazatlán, Instituto de Ciencias del Mar y Limnología-UNAM, Mazatlán, Sinaloa 82040, Mexico; 5Laboratory of Molecular and Cellular Biology, Centro de Investigación en Alimentación y Desarrollo, Mazatlán, Sinaloa 82112, Mexico

**Keywords:** congenital malformations, sea turtle embryos, epigenetic mechanisms, environmental factors, embryonic mortality

## Abstract

**Simple Summary:**

Congenital malformations can lead to embryonic mortality in many species, and sea turtles are no exception. Genetic and/or environmental alterations occur during early development in the embryo, and may produce aberrant phenotypes, many of which are incompatible with life. Causes of malformations are multifactorial; genetic factors may include mutations, chromosomal aberrations, and inbreeding effects, whereas non-genetic factors may include nutrition, hyperthermia, low moisture, radiation, and contamination. It is possible to monitor and control some of these factors (such as temperature and humidity) in nesting beaches, and toxic compounds in feeding areas, which can be transferred to the embryo through their lipophilic properties. In this review, we describe possible causes of different types of malformations observed in sea turtle embryos, as well as some actions that may help reduce embryonic mortality.

**Abstract:**

The completion of embryonic development depends, in part, on the interplay between genetic factors and environmental conditions, and any alteration during development may affect embryonic genetic and epigenetic regulatory pathways leading to congenital malformations, which are mostly incompatible with life. Oviparous reptiles, such as sea turtles, that produce numerous eggs in a clutch that is buried on the beach provide an opportunity to study embryonic mortality associated with malformations that occur at different times during development, or that prevent the hatchling from emerging from the nest. In sea turtles, the presence of congenital malformations frequently leads to mortality. A few years ago, a detailed study was performed on external congenital malformations in three species of sea turtles from the Mexican Pacific and Caribbean coasts, the hawksbill turtle, *Eretmochelys imbricata* (*n* = 23,559 eggs), the green turtle, *Chelonia mydas* (*n* = 17,690 eggs), and the olive ridley, *Lepidochelys olivacea* (*n* = 20,257 eggs), finding 63 types of congenital malformations, of which 38 were new reports. Of the three species, the olive ridley showed a higher incidence of severe anomalies in the craniofacial region (49%), indicating alterations of early developmental pathways; however, several malformations were also observed in the body, including defects in the carapace (45%) and limbs (33%), as well as pigmentation disorders (20%), indicating that deviations occurred during the middle and later stages of development. Although intrinsic factors (i.e., genetic mutations or epigenetic modifications) are difficult to monitor in the field, some environmental factors (such as the incubation temperature, humidity, and probably the status of feeding areas) are, to some extent, less difficult to monitor and/or control. In this review, we describe the aetiology of different malformations observed in sea turtle embryos, and provide some actions that can reduce embryonic mortality.

## 1. Introduction

Teratology, from the Greek *teras*, meaning monster, is the study of causes and mechanisms leading to abnormal development. Developmental malformations are multifactorial, and may include autosomal genetic diseases, point mutations, chromosome aberrations, as well as non-genetic/environmental factors, such as maternal health and nutrition, mechanical problems, exposure to toxicants, radiation or hyperthermia [1]. Teratogens may interfere with developmental processes resulting in congenital malformations; teratogenic agents may act at any time during development, and depending on the severity, the embryo may die or be malformed (Table 1).

Embryonic development in sea turtles largely depends on the external environment, and extreme environmental conditions can cause congenital malformations. Sea turtles are ectothermic animals displaying sexual reproduction with internal fertilisation. After mating, females can store viable sperm to fertilise more eggs; fertilization occurs in the upper oviduct, and although no specialized sperm storage structures are present in sea turtles, sperm is stored in the ducts of albumin glands in the upper region of the oviduct [2,3,4]; therefore the offspring of a single female may be from different males [5,6]. Fertilised eggs contain not only all the nutrients necessary for development to occur, but also accumulate xenobiotics, which can be maternally transferred because of their lipophilic properties [7]. Sea turtles search for a suitable place to nest, at which time embryonic development is arrested in middle gastrula stage; once the eggs are laid in the sand, development resumes, which will depend entirely on environmental factors, such as temperature and humidity, until hatchling emergence [5,8,9,10,11,12].

Bárcenas-Ibarra et al. [13] reported the results from an extensive field survey in which congenital malformations were recorded based on external anatomical characters of three sea turtle species, hawksbill (*n* = 23,559 eggs) and green (*n* = 17,690 eggs) from the Mexican Caribbean, and olive ridley (*n* = 20,257 eggs) from the Mexican Pacific; 63 types of congenital malformations were identified, of which 38 were novel, with frequencies between 0.2 to 2%. Some data also exist for the kemp’s ridley, reporting a frequency of 4% [14]. Craniofacial malformations were common in all sea turtle species, indicating alterations of early developmental pathways (Table 1). Malformations in the body were also observed, including carapace and limb defects, as well as pigmentation disorders, indicating alterations during later stages of development (Table 1). Of the three species, the olive ridley sea turtle presented a higher incidence of congenital malformations with a high number of severe anomalies in the craniofacial region (49%), defects in the carapace (45%) and limbs (33%), as well as pigmentation disorders (20%).

Interestingly, some alterations in DNA methylation patterns in *L. olivacea* embryos have been described related to eye defects [15] and schistosomus reflexus syndrome [16,17]. Embryos with eye defects had unique DNA methylation patterns in the putative promoter of the *pax6* gene, and some methylated cytosines were detected in transcription factor-binding sites for the aryl hydrocarbon receptor (AhR), the aryl hydrocarbon receptor-nuclear translocator complex (AhR-ARNT), the aryl hydrocarbon receptor-hypoxia induction factor (AhR-HIF), and the oestrogen receptor. In the second case, embryos with congenital malformations (with or without schistosomus reflexus) showed higher mercury concentrations in their tissues compared to normal embryos, whereas embryos with schistosomus reflexus showed higher global DNA methylation levels compared to embryos without or other malformations. These reports suggest that epigenetic mechanisms regulate interactions between genes and environment, supporting the hypothesis that malformations in sea turtles are driven (at least in part) by an environmental component.

There are seven sea turtle species in the world: leatherback (*Dermochelys coriacea*), hawksbill (*Eretmochelys imbricata*), loggerhead (*Caretta caretta*), green (*Chelonia mydas*), kemp’s ridley (*Lepidochelys kempii*), olive ridley (*Lepidochelys olivacea*), and Australian flatback (*Natator depressus*). All are considered endangered in the International Union for Conservation of Nature (IUCN) Red List with exception of *N. depressus* (considered data deficient, IUCN, 2020 https://www.iucnredlist.org/species/14363/4435952, (accessed on 5 February 2021). The difference in the incidence of malformations reported between different sea turtle species responds to different distributions, ecological niches (e.g., leatherback is a predator of jellyfish and the green turtle forages sea grasses), as well as resilience capacities [18]. For instance, it has been reported that gnathoschisis (cleft palate) is the prevalent craniofacial malformation in green, loggerhead, and olive ridley embryos [13,19]; while variation in scute patterns is the most common malformation in hard-shelled sea turtles [13,20], and leucism (hypopigmentation) has been reported in several species of turtles [13,21,22,23,24,25]. Thus, in this review, we describe some of the most common malformations observed in sea turtle embryos and hatchlings, possible causes, and some actions to reduce embryonic mortality due to abnormal development.

## 2. Pigmentation Disorders

Leucism (*leuc*- is the Latin variant of *leuk*- from the Greek *leukos* meaning “white”) is a congenital condition showing discoloration of certain parts of the body (hypopigmentation) related to disorders in pigment cell differentiation and/or migration during development producing chromatic aberrations [32]. In leucism, all pigment-producing cells differentiate from the same precursor cell, and therefore it causes a reduction in all types of pigments. Contrary to albinism, in which one or several genes that produce or distribute melanin are defective (showing a complete lack of pigmentation including eyes, skin, and nails), in leucism normal coloration is observed in some parts of the body (such as eyes and nails) [33,34]. Leucism has commonly been referred to as partial albinism, and has been associated with several non-mutually exclusive factors such as pollution [32,35], diet [33], follicular damage [36], genetic mutations [32], or age and sex [32,34]. The lack of pigmentation has also been associated with inbreeding, in which recessive alleles would be expressed [37], indicating low levels of genetic diversity [33]. A leucistic or albino condition may affect fitness, since coloration is an important component for survival (i.e., predator detection) and reproduction (i.e., mate selection) [35].

Leucism has been observed in sea turtle embryos at later stages (sometimes together with other malformations) (Figure 1) [13,38], and they can even reach adulthood [31]. Leucism, when not accompanied by other malformations, can be compatible with life, since the first pigments appear during early growth [11] and total pigmentation appears during late growth when the main characteristic is exponential growth in mass [8], until hatching (Table 1). A possible aetiology could be related to low quality maternal nutrition or exposure to chemical compounds present in the food chain or the water, altering (either by mutational or epigenetic mechanisms) the expression of pigment-producing genes. More than 150 pigmentation-related genes have been identified, harbouring different types of mutations such as single-nucleotide polymorphisms, insertions, deletions, or duplications [38].

Interestingly, multiple leucism/albinism has been reported in the loggerhead sea turtle *Caretta caretta* (22.4% of the nest) [22,24] and the yellow-spotted Amazon River turtle *Podocnemis unifilis* (43.24% of the nest) [24], suggesting inbreeding as a potential cause, since leucistic individuals were commonly found in small and isolated populations; this isolation could be caused by human activities, leading to a reduction in gene flow [24]. Recently, Madeira et al. [25] reported a clutch of 30 out of 123 (24.4%) albino green sea turtle hatchlings in Poilão Island, Bijagós Archipelago, Guinea-Bissau. The authors reported a normally pigmented female as a mother, but no genetic information about paternity was recorded to know if these organisms originate from the same father. Noteworthy, the body mass and size of leucistic or albino hatchlings were greater than their normal siblings [23,24,25], and this could be an advantage to compensate for the lack of pigments, since these individuals can survive to adulthood, remain reproductively active, and contribute to the population [31]. More studies are needed to understand these differences in size and weight.

## 3. Craniofacial Disorders

According to previous reports concerning green, loggerhead, and olive ridley sea turtles, craniofacial malformations are the most prevalent [13,19,39]. Craniofacial and brain development are complex interconnected processes in which cephalic neural crest cells are the most important contributors of tissues and sensorial organs [40]. Important signalling regulators for craniofacial development include Sonic hedgehog (Shh) and Wnt, plus families of growth factors (such as fibroblast growth factor (FGF), transforming growth factor (TGF), or epidermic growth factor (EGF)) and retinoic acid (RA), which regulate the interaction of cephalic neural crest cells with other cells; thus gene mutations, chromosomal aberrations, and/or environmental factors affecting these signalling pathways may result in congenital malformations [41].

### 3.1. Head

The head is a complex structure derived from endoderm, mesoderm, ectoderm, and cephalic neural crest cells. The morphogenetic process involves hundreds of genes interacting to regulate cell proliferation, migration, and differentiation in a coordinated fashion. These regulatory processes are highly dynamic and susceptible to dysregulation [42], illustrated by the large amount of cranial malformations documented in animals, some of which have been observed in sea turtles.

Bicephaly refers to the presence of two heads in a single individual, and has been documented in sea turtles (Figure 2) [13]. According to Velo-Antón et al. [43], the term is used to describe a broad spectrum of developmental alterations, such as the duplication of head structures or incomplete splitting of the zygote related to the occurrence of conjoined twins (which have also been documented in sea turtles, Figure 3). In addition, bicephaly may occur by terminal bifurcation of the notochord during neurulation [42]. Sea turtles may hatch with this condition but do not live long (hours or days) and the causes are unknown, presumably genetic, environmental or a combination of both [44]. Interestingly, bicephaly is by far the most reported malformation in snakes, both wild and in captivity, and its associated causes could be both cool and hot temperatures during incubation, inbreeding depression, hybridization, environmental pollution, and chemical toxins [44].

Microcephaly has also been observed in sea turtle embryos [13] and, according to Pirozzi et al. [45], this is a congenital malformation associated with deficient brain growth, which is determined by the orchestration of tightly regulated processes involving cell proliferation, migration, and death. Mutations (deletions or duplications) in several human genes, defective DNA-repair pathways, as well as chromosome aberrations have been linked to these critical processes impacting brain development [46]. Nevertheless, non-genetic factors, such as environmental conditions, viral infections, and exposure to toxic compounds, may also result in a reduction in brain size [45]. For instance, in humans, specific genetic mutations and some viral infections (i.e., Zika) have been linked to microcephaly [46,47]; however, to our knowledge, no aetiological agents associated with microcephaly have been documented in sea turtles. Because developmental processes are usually conserved, it is possible that similar genetic and non-genetic factors disrupting brain development in humans could also affect brain growth in sea turtles.

Neural tube defects have also been observed in sea turtle embryos [13]; these defects are also multifactorial [48]. Anencephaly occurs during embryogenesis when the neural tube fails to close completely; this condition results from a failed closure of the neural folds in the mid and hindbrain, leading first to exencephaly (also observed in sea turtles), in which the neuroepithelium is exposed, bulging from the brain. This tissue further degenerates, resulting in anencephaly [48]. Encephalocele is another neural tube defect documented in sea turtle embryos; it refers to sac-like protrusions of the brain and its membranes (meninges) caused by failure of the neural tube to close completely. In humans, the genetic basis of neural tube defects is well described, identifying folic acid deficiency as a major causative agent [48,49,50], suggesting nutritional causes. Other non-genetic factors causing neural tube defects such as infections, drugs, caffeine, smoking, and alcohol have been identified in humans, as well as hyperthermia [49]; the latter is relevant in sea turtles since the incubation temperature is crucial during development, and sustained high temperatures may cause congenital malformations. Nevertheless, the effects of hyperthermia on neural tube defects are still inconclusive, since one study did not find evidence that high temperature during sea turtle development causes neural tube defects [29].

### 3.2. Forebrain and Midline Facial Structures

The most severe phenotypes in the midline facial structure (such as cyclopia, arhinia, and agnathia) identified in sea turtles occur during early development, and could be intrinsically related to the absence or incomplete division of the forebrain into two cerebral hemispheres (i.e., holoprosencephaly), which can also result in mild malformations [51]. Cyclopia has been reported in the olive ridley [13,19]; it is the most severe ocular malformation characterized by the presence of a single eye in the middle of the face due to the presence of a single optic vesicle, and it is always accompanied by multiple craniofacial malformations caused by holoprosencephaly [52]. Arhinia is the absence of nares, and it has been reported in the green turtle [13]. A leatherback sea turtle embryo with holoprosencephaly, arhinia, proboscis, and maxillary micrognathia has been reported, and according to the authors, these malformations could be related to the nutritional status of the mother and/or exposure to mercury [53]. Agnathia (Figure 1) is the absence of jaws (maxilla and/or mandible); it has been identified (although in low proportions) in green, hawksbill, and olive ridley sea turtle embryos [13,19], and it is closely linked to holoprosencephaly [54].

Shh signalling plays a major role in forebrain division, and a loss-of-function of this pathway leads to severe aberrant phenotypes in the mid-craniofacial region [52]. Shh, together with other cell signalling networks (i.e., bone morphogenetic protein (BMP), FGF, Nodal, and RA), controls morphogenetic events for this process and facial development [55]. The proteins of the Hedgehog (HH) family must undergo two post-translational modifications so that they can be secreted as active signalling molecules, first by a cholesterol residue at their N-terminal end, and then by a palmitate residue at their C-terminal end [56,57]. A blockage of Shh signalling in the brain would inhibit the subsequent induction of *Shh* expression in the frontonasal ectodermal zone (FEZ), a signalling centre that regulates facial development [58]. It is possible that the Shh signalling pathway largely depends on the nutritional status of the mother, with effects on both brain and face development. A deficiency of Shh (i.e., haploinsufficiency) could interact with other genetic and/or epigenetic factors, leading to holoprosencephaly [59], producing aberrant phenotypes with varying degrees of severity. Interestingly, lower blood cholesterol levels have been reported for green turtles with fibropapilloma (as well as with other pathologies) compared to healthy turtles [60,61,62]. Blood cholesterol has been recently proposed as a biomarker of fibropapillomatosis in green turtles, since cholesterol values are significantly lower when the pathology is more severe [63].

RA signalling converges with Shh signalling [64] and coordinates morphogenetic events during brain and face development. RA is an active derivative of vitamin A and is provided exclusively through the diet. The biogenesis of all-trans RA is the first developmental step in the initiation of RA-regulated signalling pathways to subsequently bind to nuclear RA receptors and act as transcription factors [65]. Excess or deficiency of vitamin A is detrimental to normal development [66,67]. The transforming growth factor-beta-induced factor (TGIF) is key in the regulation of RA signalling and is essential to correctly model the forebrain; it regulates the expression of genes controlling both RA synthesis and degradation [68]. Eggs of sea turtles (and of reptiles in general) have a retinoid storage mechanism to prevent hypovitaminosis A [69]. Coincidentally, lower levels of retinol have been reported in diseased green turtles compared to healthy turtles [61,70]. Furthermore, retinoid levels may vary between colonies or populations; in green turtles, significantly different concentrations of vitamin A in plasma have been reported between sampling sites, which was attributed to the availability of food [70]; in loggerhead turtles, it has been reported that plasma concentrations of vitamin A decreased as the nesting season progressed [70], which could indicate maternal transfer to eggs.

Hypoxia-ischemia has also been associated with holoprosencephaly with severe malformations in the midface (cyclopia, arhinia, and micrognathia) [71]. This is relevant in sea turtles since they depend on the exchange of oxygen through the chorion and its availability within the nest. When the forebrain separates to organise the facial structure, the cells of the neural crest begin to migrate, shortly after the closure of the neural tube [72]. This is a population of migratory cells derived from the superficial and neural ectoderm that will give rise to the primordia of the facial skeleton [73]; these cells must undergo an epithelial to mesenchymal transition (EMT) and subsequently migrate to their destinations; FoxD3 and Sox10 are necessary to initiate this process [74], while Sox9 is required for the determination of the chondrogenic lineage [75]. Coincidentally, RA inhibits *Sox9* expression [76], while hypoxia induces its expression [77].

Studies in sea turtles indicate that the jaw is the most affected craniofacial region [13,19,39], and in green, loggerhead, and olive ridley turtle embryos, gnathoschisis (cleft palate) is the predominant malformation [13,39]. The FEZ regulates maxillary growth; *Shh* is expressed in the ectoderm that will form the roof of the mouth and forms a boundary with cells expressing *Fgf8* and *Wnt9b* in the dorsal ectoderm, promoting growth and patterning of the maxilla [78]. Gnathoschisis has its origin in early embryonic development (Table 1) and is caused by an incorrect fusion of the medial nasal and maxillary prominences and frontonsal mass deriving from mesenchymal and neural crest cells [26]. Endochondral bones of the face fail to develop, resulting in multiple defects such as gnathoschisis or brachygnathia, which may be promoted by *Sox9* dysregulation [79]. Brachygnathia is characterized by shortening of the jaw (maxillary or mandibular) and has been identified in the green sea turtle as recurrent as gnathoschisis [13].

It appears that the spatial organization of gene expression patterns in FEZ is highly associated with the facial shape [80], and changes in the expression domains of signalling molecules such as *Shh*, *Fgf8, Wnt9b, Bmp2, Bmp4*, and *Bmp7* could result in a variety of aberrant phenotypes of the jaw and the nasal region in sea turtle embryos. For instance, prognathia and agnathia, which refer to the lengthening and absence, respectively, of the jaw, maxillary or mandibular, have been recurrently identified in olive ridley turtle embryos (Figure 1) [13,19]; laterognathia (characterized by lower jaw turned to right or left) has been reported in live hawksbill embryos, although at low frequencies [13]; and prosoposchisis (a fissure of the face extending from the jaw to the eye) has also been reported in olive ridley turtle embryos [13].

### 3.3. Eyes

Eye development begins with the division of the ocular field into two optic vesicles [52]. The *Pax6* gene, as part of a set of genes that encode the highly conserved ocular field transcription factor among vertebrates [81], is considered an eye selector gene necessary to initiate the regulatory cascade for lens and eye development [82]. Hence, a loss-of function on *Pax6* can cause anophthalmia (Figure 4A) [83]. After cyclopia, it is the most severe phenotype characterized by absence of the eyes. In green turtle embryos, anophthalmia has been reported as prevalent, above gnathoschisis [13].

Pax6 together with Rax, Six3 and Lhx2 constitute a group of eye field transcription factors (EFTFs) necessary for ocular development [81]. Sox2 and Otx2 activate the expression of *Rax*, inducing up-regulation of the EFTFs. In olive ridley sea turtle embryos, microphthalmia (abnormally small eyes) has been reported, although in very low frequencies [13]. In humans, Sox2 and Otx2 are both the major causative factors for anophthalmia and severe microphthalmia [84].

The RA signalling pathway could be involved in ocular malformations, since severe ocular abnormalities, including microphthalmia, have been observed in mice whose mothers were treated with RA [85]. Paradoxically, a loss of function of the *Stra6* gene, a cellular receptor that binds to the retinol-binding protein and plays an important role in the transport of retinol to the eye [86], is also a cause of microphthalmia and anophthalmia in humans [87]. These reports suggest a nutritional aetiology; however, in addition to nutritional factors, pollutants and gas exchange in the embryo may contribute to craniofacial malformations. Variations in temperature and humidity can also generate aberrant phenotypes [11,88,89], although the mechanisms of action are not totally understood.

Macrophthalmia, characterized by abnormally big eyes, has also been reported in olive ridley embryos [13]. In humans and mice, haploinsufficiency of CRIM1 (cysteine-rich transmembrane bone morphogenetic protein (BMP) regulator 1) has been reported as the cause of macrophthalmia [90]. Interestingly, contaminants such as polycyclic aromatic hydrocarbons (PAHs) can induce transcriptional changes on CRIM1 and disrupt multiple cellular processes [91]. Synophthalmia, also reported for olive ridley turtle embryos [13], is characterized by eyes more or less completely fused into one, and is related to the incomplete division of the optic field.

### 3.4. Nares

From the FEZ, two nasal placodes emerge, which are then invaded by growing ectoderm and mesenchyme, and then fuse to become the nasal cavity and the primitive choana. The expression of *Fgf8* is required for proper morphogenesis of the entire nasal cavity [92], and recently, SMCHD1 (structural maintenance of chromosomes flexible hinge-domain containing protein 1) has emerged as a key regulator of embryonic genomic function as an epigenetic modifier in several genes [93]. In humans, mutations in the *SMCHD1* gene have been identified as causing arhinia with different degrees of severity, and Bosma Arhinia Microphthalmia Syndrome [94].

In sea turtle embryos, malformations in the nares are less common compared to malformations of the jaw and eyes [13,19,39]. Rinoschisis, a peculiar and rare malformation, characterized by vertical separation of the nares into two equal parts, has been identified in hawksbill turtle embryos [13]; in humans, mutations in the *FREM1* gene (FRAS1-related extracellular matrix protein 1) have been reported to cause this malformation [95]. Another rare malformation is rhinocephaly, consisting of one proboscis-shaped nare generally in the front of the head; it has been reported for olive ridley and leatherback turtle embryos [13,53] related to holoprosencephaly.

## 4. Skeletal Defects

Bones, together with joints and cartilage, are part of the skeleton in all vertebrates. Skeletogenesis starts during embryonic development when multipotent mesenchymal stem cells migrate from mesoderm and ectoderm to specific sites of the body and commit to the skeletal fate. Initially, the primary skeleton is cartilaginous, and then it transforms into bone by a process known as endochordal ossification [96].

### 4.1. Spine

Spinal defects (such as kyphosis and scoliosis) have been observed in sea turtles. Kyphosis (Figure 5) is a spinal disorder showing an excessive outward curve or a convex deformity of the spine in the sagittal plane, resulting in an abnormal rounding of the upper back. The condition is sometimes known as roundback or hunchback, and it has been reported in different turtle species [13,19,97,98,99]. Scoliosis, however, refers to a lateral curvature of the spine in the frontal plane, and this condition has been rarely reported in turtles [13,99]. Both conditions may occur together (kyphoscoliosis), as reported by Turner [99] in the snapping turtle *Elseya irwini*. According to the author, these spinal deformities may not significantly reduce individual fitness, but could be associated with other malformations or internal abnormalities.

Animal models have been used to understand the causes of spinal defects, and although aetiologies are still elusive, several factors such as mechanical, genetic predisposition, errors during vertebral segmentation/ossification, neuroendocrine dysfunction, and nutritional or traumatic events have been implicated, suggesting a multifactorial process [100].

### 4.2. Bones

The failure of the bones to develop during embryonic growth is known as bone agenesis, and the specific terminology depends on the type of missing bone or structure. For instance, the absence of hands and/or feet is called meromelia, the absence of long bones is known as phocomelia, the absence of one or more limbs is called amelia (Figure 4B), shorter or smaller limbs is known as micromelia, partial absence of a limb is termed hemimelia, and the absence of the tail is called anury; most of these disorders have been documented in sea turtles [13].

In humans, twin and family studies indicate that bone development is regulated by several genes (polygenic), and that hormones and environmental factors play important roles. Other studies in humans and animal models have shown that mother undernutrition (such as low calcium and vitamin D intake) may alter gene expression during bone development in the foetus, and this event could be a critical determinant later in life [101]. This epigenetic effect may become transgenerational, and bone defects may be observed in subsequent generations [101]. Heritability also seems to be an important factor accounting for bone defects. It is suggested that mutations in both regulatory and coding gene sequences accumulate producing morphological differences between and within species [96]; some of these mutations could disrupt skeletogenesis, leading to bone malformations. Considering that developmental processes are conserved among vertebrates, it is likely that the same mechanisms disrupting skeletal development in humans and animal models operate in turtles, and bone defects would also be multifactorial in sea turtle species, with an important environmental influence.

### 4.3. Limbs and Digits

Limb and digit development depends on two signalling regions: the apical ectodermal ridge (AER) and the zone of polarising activity (ZPA). Animal models such as chicken and mouse embryos have been used to understand limb development (growth and patterning); manipulations of the limb bud during embryonic development identified the AER as necessary for the proximal-distal (shoulder to digits) outgrowth of the limb, whereas the ZPA is essential for the anterior-posterior patterning of the limb (digit thumb-to-pinkie development) through morphogenic action [27,28]. Both regulatory foci are interrelated to form a normal limb. AER removal results in the termination of limb bud outgrowth, producing a truncated proximal–distal axis, whereas transplantation of the ZPA into the anterior mesoderm of another limb induces mirror-image duplications of the anterior-posterior axis of the limb, and the duplicated limb also presents a correct proximal-distal axis [28]. This process largely depends on FGF signalling. *Shh* is expressed in the ZPA and can induce a mirror image of the anterior-posterior axis if introduced to the anterior mesoderm of another limb bud; Shh activates *Ffg4* transcription in the limb ectoderm, and *Bmp2* and *HoxD11* in the limb mesoderm; moreover, FGF and RA are necessary for the expression of *Shh* as a positive feedback loop [27,28]. In fact, *Shh* expression in the limb bud is regulated by the zone of polarising activity regulatory sequence (ZRS, a limb-specific enhancer of *Shh*), which contains different domains [102,103,104]. Interestingly, *Shh* expression levels are controlled by overlapping domains containing HoxD binding sites, and gene expression depends on the number of binding sites [104]. HoxD belongs to a family of homeodomain-containing transcription factors necessary for the specification of body plans along the anterior-posterior axis in animals [105], and differential expression and timing of *Hox* genes in the lateral plate mesoderm have been implicated in the correct limb placement through RA signalling [106], evincing the complexity (and vulnerability) of regulatory networks involved in limb and digit development.

Digits differentiate as radial condensations within the autopod (the distal paddle-shaped region of the limb bud), which contains mesodermal-derived skeletal progenitors covered by ectoderm. Autopod outgrowth is directed by the AER, which provides growth factors to activate proliferation of mesodermal progenitors, which condensate to form radial cartilage blastemas and become segmented into phalanges; at the same time, progenitors in the interdigital regions remain undifferentiated and may undergo cell death to different extents depending on the species; these progenitor cells also retain a potential to develop extra digits [107]. Disruption of digit development and/or interdigital remodelling may lead to digit defects such as polydactyly or syndactyly [107,108]. In mammals (including humans), a condition termed synpolydatyly is known as an autosomal dominant limb malformation in which exon1 from the *HoxD13* gene contains an imperfect trinucleotide repeat sequence encoding for a 15-residue polyalanine tract, which destabilizes the protein, interfering with protein translocation from the cytoplasm to the nucleus [108]. Variations in *HoxD13* mutations have been identified to cause overlapping conditions such as brachydactyly, syndactyly, and brachydactyly–syndactyly syndrome, among others, and therefore the term “HoxD13 limb morphopaties” has been adopted to encompass all HoxD13-related malformations [109]. Other signalling pathways such as BMP-, FGF-, Wnt-, all-trans-RA, and Notch-signalling have been implicated in digit development [107].

Malformations such as polydactyly, brachydactyly, and syndactyly have been documented in sea turtles [13,19] (Figure 6), and whether the same mutations in *HoxD13* (and related signalling pathways) disrupt digit development in these species requires further research. Interestingly, epigenetic regulation of *Hox* genes through the Trithorax and Polycomb groups has been documented [110], highlighting the effects of the environment in limb and digit development, and presenting an interesting area of research in comparative teratology.

## 5. Carapace Defects

The turtle shell evolved from endoskeleton modifications, in which costal and neural plates are hypertrophied ribs and vertebrae, respectively [111]. It consists of two main parts: the dorsal carapace and the ventral plastron, connected by lateral bridges. Turtle shell development begins with the formation of a specialized structure, the carapacial ridge [112,113,114], which is a protuberance of mesenchymal cells covered by ectoderm located in the thoracic region; the carapacial primordium expands laterally, followed by rib primordia, causing dorsalisation of the ribs. The ribs are then covered by the dorsal dermis and, instead of extending ventrally the ribs, integrate to the carapacial dermis [114,115]. The shell is covered by an array of modular epidermal structures called scutes that develop from primary signalling centres along the carapacial ridge [112]. Thus, fully developed shells are composed of ribs, vertebrae, clavicles, and scutes. Interestingly, the inhibition of Shh and BMP signalling pathways can suppress scute formation, highlighting the importance of signalling centres located at the carapacial ridge [115].

Sea turtle embryos and hatchlings present different carapacial anomalies, such as a compressed carapace, misshapen, supernumerary or subnumerary scutes [13], and according to Velo-Antón et al. [43], the presence of anomalies, malformations, and asymmetries in wild animals could be an indicator of developmental instability. Variation in scute patterns is the most common malformation in the carapace of hard-shelled sea turtles, but there is no evidence that these variations affect survival [11,13,20,116]. The main proven causes of these anomalies are hot temperatures and dry conditions [29,30], although it has also been suggested that the presence of metals could be involved [117]. In addition, crowding pressure on the eggs inside the nest may contribute to carapacial deformities [13].

Zimm et al. [29] analysed data from three sea turtle natural nesting conditions under the hypothesis that scutes were patterned by reaction-diffusion dynamics, which could explain mechanisms of anomalous variation, finding that environmental factors (mostly temperature and humidity) play a key role in intra-species scute variations. This was consistent with a previous study from Telemeco et al. [30] reporting the influence of extreme hot temperatures and dry incubation conditions for more than 60 h on the incidence of scute anomalies. These variations did not seem to be heritable, but emerged from the dynamics of development under the influence of the environment; apparently mid-developmental stages were more susceptible to environmental conditions, leading to scute anomalies [29], because it is the time when the scales begin to be visible [11] (Table 1).

## 6. Schistosomus Reflexus Syndrome

Bárcenas-Ibarra et al. [16] reported the presence of schistosomus reflexus syndrome in olive ridley sea turtle embryos. This syndrome is a rare, fatal congenital malformation occurring in rumiants, mostly in cattle. It was identified in 0.6% of all embryos, and in 31% of the embryos with congenital malformations over a 7-month period. The main characteristics of this syndrome are the lack of closure in the abdominal wall, spinal inversion, limb ankylosis, positioning of the limbs adjacent to the skull, and lung and diaphragm hypoplasia; other features may include scoliosis and the exposure of thoracic viscera (Figure 4B) [16,118].

For a long time, the aetiology of schistosomus reflexus has been thought to be genetic [118], and some hypotheses include superfetation, inbreeding, and chromosomal aberrations [119,120,121]; however, some evidence indicates that environmental factors may also be involved. Martín-del-Campo et al. [17] reported higher mercury concentrations in olive ridley sea turtle embryos (with or without schistosomus reflexus) compared to normal embryos, and a positive correlation between mercury concentrations and DNA methylation in embryos with schistosomus reflexus. This suggests that environmental contaminants (such as mercury) and epigenetic mechanisms (such as alterations in DNA methylation levels) could be among the causative agents. More research is necessary to fully understand the effects of the environment in the aetiology of this complex malformative syndrome.

## 7. Discussion

Congenital malformations originate from multifactorial aetiologies, involving a complex interplay between genetic, epigenetic, and environmental factors influencing developmental programmes, affecting transcriptional networks and protein function and resulting in aberrant phenotypes [122]. Depending on the time and duration of exposure, even subtle genetic or epigenetic modifications may promote dramatic alterations of developmental processes, resulting in physical and/or physiological malformations, usually incompatible with life [15,122].

Several environmental factors including temperature, radiation, pH, salinity, humidity, infections, and chemical compounds are potentially teratogenic. Nevertheless, we will focus this discussion on temperature, humidity, and anthropogenic-derived contaminants (i.e., persistent organic compounds such as pesticides and hydrocarbons, widely distributed in the aquatic environment) because these factors have been monitored in both feeding and nesting areas, providing useful information to help improve management.

Hyperthermia has been regarded as teratogenic in different mammalian species [123,124,125]. The teratogenic effect of extreme temperatures depends on timing, intensity, duration, and actual temperature involved [123]. The incidence of congenital malformations depends on the species and the embryological stage at the time of exposure [121,122]. In most mammalian species, hyperthermia is reached between 1.5 and 2.5 °C above the normal body temperature; in fact, 1.5 °C above normal temperature in the pre-implantation embryo may result in a high mortality rate, whereas after implantation it may cause congenital malformations [126]. Temperature-associated malformations include neural tube defects, microphtalmia, microcephaly, limb defects, craniofacial malformations, and developmental defects of internal organs such as the abdominal wall, heart, and kidneys [123].

In ectothermic animals, incubation temperatures during embryonic development affect different traits such as body size and shape, locomotor performance, and sexual phenotype, among others [127,128]. Exposure to high temperatures can result in neural tube defects and craniofacial malformations, plus defects in limbs and vertebrae [129,130,131]. In freshwater turtles, malformations were observed in the tail and carapace [132]. As in mammals, early developmental stages in reptiles are more susceptible to the teratogenic effects of high temperatures [128,131,132]. In sea turtles, hyperthermia and low humidity during embryonic development may cause low hatching success, high mortality and female-biased sex ratios [133,134,135,136], as well as craniofacial, limb, and pigmentation disorders [21]. Scute anomalies are the best-characterised malformations documented in turtles exposed to high temperatures. Telemeco et al. [30] examined the effects of temperature during the embryonic development of painted turtles (*Chrysemys picta*) in natural nests, reporting that exposure to extreme hot temperatures and dry conditions for more than 60 h produced more carapace abnormalities than shorter exposure times or colder temperatures, and these carapacial malformations occurred regardless of carapacial deformities in the mother. The effect of temperature in scute variations was also confirmed by Navarro-Sánchez [137] and Zimm et al. [29] in sea turtles, and although scute variations are compatible with life, it is unclear whether they affect fitness.

Anthropogenic contamination, such as pesticides and polychlorinated biphenyls (PCBs), has also been implicated in abnormal development. Pesticides are classified depending on target species as insecticides, herbicides, nematicides, piscicides, molluscicides, avicides, rodenticides, bactericides, and fungicides; they are toxic and bioaccumulate in the food chain [138]. While only a small fraction of applied pesticides reach target pests, most are dispersed in the environment, reaching non-target species. Specific toxic effects and the persistence of different groups of pesticides have been recently reviewed [138], including their effects on oxidative stress, metabolic alterations, endocrine disruption, and developmental anomalies [138,139].

Studies with mammals, birds, amphibians, and fish have shown that experimental exposure to pesticides (such as aldrin, dieldrin, endrin chlorpyrifos, diazinon, alachlor, atrazine, and deltamethrin) causes several types of malformations, mostly craniofacial anomalies (including cleft palate, micrognathia, microphthalmia, exophthalmia, anophthalmia, nasal defects, plagiocephaly, exencephaly, microcephaly, and hydrocephaly), limb (syndactyly) and spinal (lordosis, scoliosis, kyphosis, and tail winding) defects [139,140,141,142,143,144].

In reptiles, most reports regarding congenital malformations have been documented in wild populations; in these studies, the eggs or embryos were analysed for the presence of organic compounds or heavy metals to establish a possible association between malformations and the presence/concentration of chemical compounds. Bishop et al. [145] reported carapace, tail, limb, and craniofacial malformations in *Chelydra serpentina*, associated with the presence of PCBs, polychlorinated dibenzodioxins (PCDDs), and polychlorinated dibenzofurans (PCDFs) in the eggs. Later, Bell et al. [146] reported tail, carapace, plastron, neural tube, and craniofacial malformations in *C. serpentina* and *C. picta*, probably associated with the presence of PAHs, suggesting maternal transfer, since lipid analysis from female turtles revealed high concentrations of PAHs. This is important because persistent organic compounds are lipophilic, and maternal transfer of these compounds during early development may alter epigenetic reprogramming by modifying DNA methylation dynamics during embryogenesis, resulting in congenital malformations [147]. In addition, PAHs can disrupt chondrocyte proliferation and reduce the expression of *Sox9* in the craniofacial skeleton, causing malformations [148].

Sea turtles also accumulate heavy metals in their tissues and these can be measured in eggs and shell fragments as a non-invasive approach [149,150,151]. Mercury has been associated with low hatching and emergence success [152]. We have recently detected mercury and pesticides (heptachlor, endosulfan, and dichlorodiphenyldichloroethylene (DDE)) in tissues from normal and malformed olive ridley embryos [17,153]. As mentioned above, mercury concentrations were higher in malformed embryos, showing association with DNA methylation in embryos with schistosomus reflexus, whereas endosulfan was detected only in malformed embryos with multiple anomalies; these embryos were siblings from a nest with a high incidence of malformations (25%), highlighting the importance of maternal feeding, maternal transfer of lipophilic compounds to the egg, and xenobiotic-mediated epigenetic alterations in the occurrence of congenital malformations.

## 8. Concluding Remarks

Embryonic mortality of sea turtles caused by congenital malformations has been well documented, and it can be reduced by implementing some actions, such as modulating the incubation temperature and humidity on nesting beaches, monitoring chemical compounds (such as organic persistent compounds and heavy metals) in feeding areas, and regulating human activities in breeding areas.

In order to mitigate the impacts of high temperatures and dry conditions, some management strategies have been proposed, such as the use of shade structures and sprinkling water on nesting hatcheries [154,155]. Shade covers are used on turtle nesting beaches to lower environmental temperature and equilibrate sex ratios [156,157,158,159]; also, sprinkling water during the night [155] or in drought conditions [158] may help maintain adequate humidity; however, it is necessary to determine the amount of shading and/or water sprinkling necessary for specific natural beach conditions.

Besides temperature, optimal embryonic development in sea turtles is influenced by humidity [133,160,161]. The monitoring of humidity on nesting beaches has received less attention than temperature, but it is also important, not only because it has been associated with the presence of abnormalities in the carapace [29], but also because it has effects on sex ratios [161]. In addition, moisture can influence hatchling size and growth by creating a more favourable environment for tissue differentiation, and it has been positively correlated with hatching success [162,163].

In Mexico, important efforts are devoted to the conservation of sea turtles. One example is Playa Espíritu (FONATUR), a protected area located in the northwest region, in the state of Sinaloa (Appendix A), where nests are relocated to protect them from extreme temperatures and predators. Temperature is monitored and controlled by temperature sensors and shade structures, respectively; humidity is managed by a sand sanitisation procedure involving sand screening, aeration, flattening, and water sprinkling (Appendix A), resulting in consistent survival and hatching rates above 85% (Appendix A). Thus, monitoring and regulating temperature and humidity in nesting hatcheries is important because these environmental variables seem to be more related to the presence of malformations and to the increase in embryonic mortality.

Feeding areas are also important because persistent organic compounds are lipophilic and bioaccumulate in the food chain. During vitellogenesis, these lipophilic compounds are transferred from the mother’s tissues to the eggs and then to the embryo, potentially altering early development [7,152]. Higher levels of contaminants (i.e., persistent organic pollutants) in plasma have been identified in sea turtle aggregations foraging in heavily urbanized coastlines compared to aggregations foraging in National Wildlife Refuges [164]. Not surprisingly, the Gulf of California that supports high productivity and biodiversity, and where several species of sea turtles converge to feed [165], shows contamination from different anthropogenic sources. According to Paez-Osuna et al. [166], ecosystems in the Gulf of California receive contamination from mining spills, intensive agricultural and shrimp farming, and harmful algal blooms; yet, the pollution levels remain relatively low to moderate depending on the location and type of contaminant(s). It is recommended to establish monitoring programmes in sea turtle feeding areas to measure relevant contaminants depending on anthropogenic activities in the area (agriculture, mining, tourism, industry, etc.). Several contaminants are endocrine disruptors and/or teratogenic, and could affect the reproduction and survival of sea turtle populations [138,143,145,146,147]. Based on results from monitoring programmes, regulations should be implemented to reduce human activities in feeding and breeding areas to lower the presence of possible anthropogenic teratogens in marine ecosystems that could eventually be transferred to the embryos, and to avoid the formation of endogamic populations that may express abnormal phenotypes.

## Figures and Tables

**Figure 1 animals-11-00444-f001:**
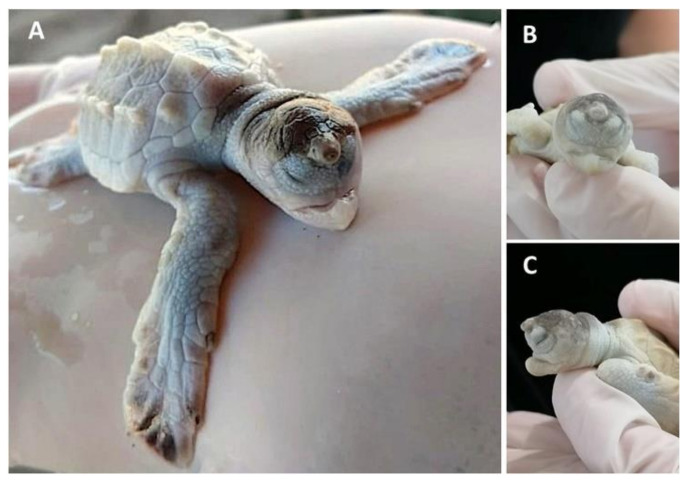
*Lepidochelys olivacea* embryo with frontal proboscis, orbital hypotelorism (decrease in the distance between the two eyes), leucism (lack of pigmentation), and maxillar agnathia (lack of upper jaw). (**A**) Whole body view; (**B**) frontal view; (**C**) side view. Photographs provided by María Fernanda Calderón-Campuzano.

**Figure 2 animals-11-00444-f002:**
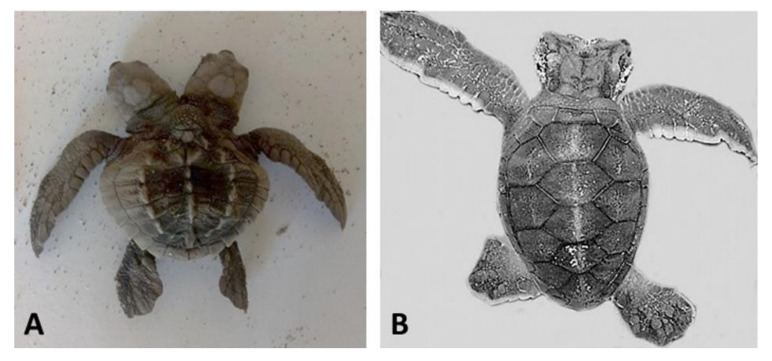
Sea turtle embryos with bicephaly (the presence of two heads). (**A**) *Lepidochelys olivacea*; (**B**) *Chelonia mydas*. Photographs provided by María Fernanda Calderón-Campuzano (**A**) and Julieta Álvarez-Servín (**B**).

**Figure 3 animals-11-00444-f003:**
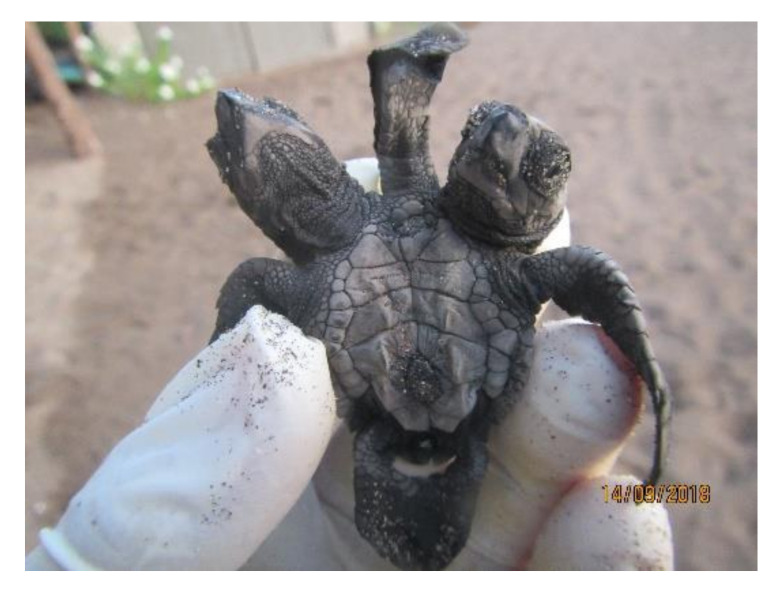
Conjoined twins (*Lepidochelys olivacea*). Photograph provided by María Fernanda Calderón-Campuzano.

**Figure 4 animals-11-00444-f004:**
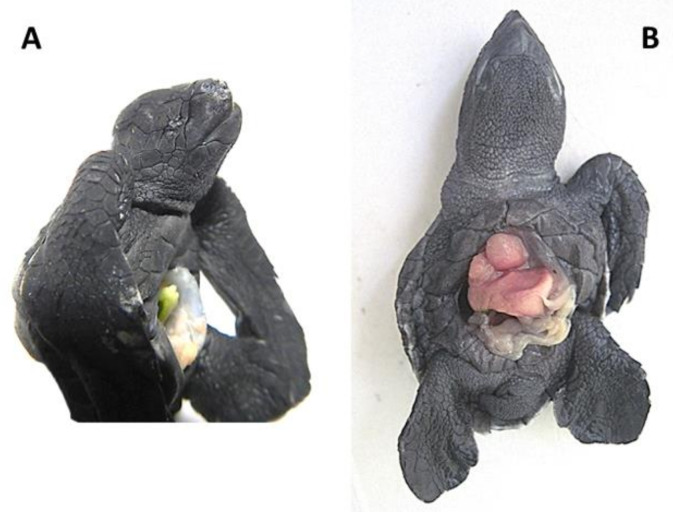
*Lepidochelys olivacea* embryos with (**A**) right anophthalmia (absence of the right eye,) and (**B**) amelia (absence of flipper) and schistosomus reflexus syndrome (ventral view). Photographs taken from supplementary file in [13] with the permission of the authors and publisher (John Wiley & Sons).

**Figure 5 animals-11-00444-f005:**
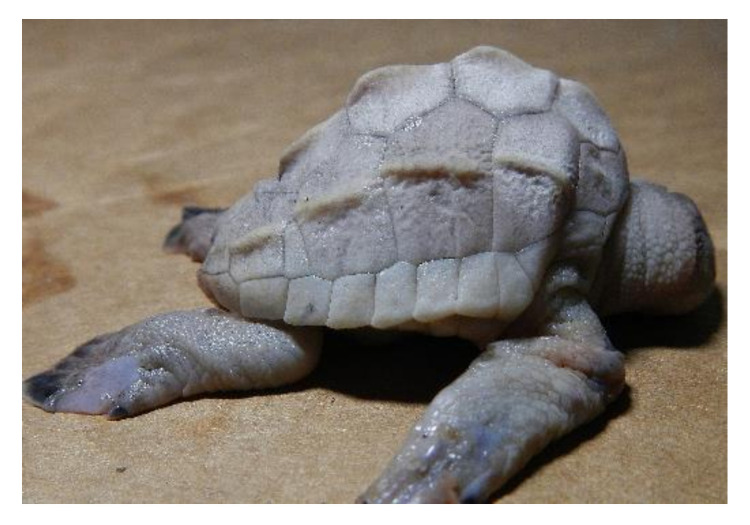
*Lepidochelys olivacea* showing kyphosis (right side view). Photograph taken from supplementary file in [13] with the permission of the authors and publisher (John Wiley & Sons).

**Figure 6 animals-11-00444-f006:**
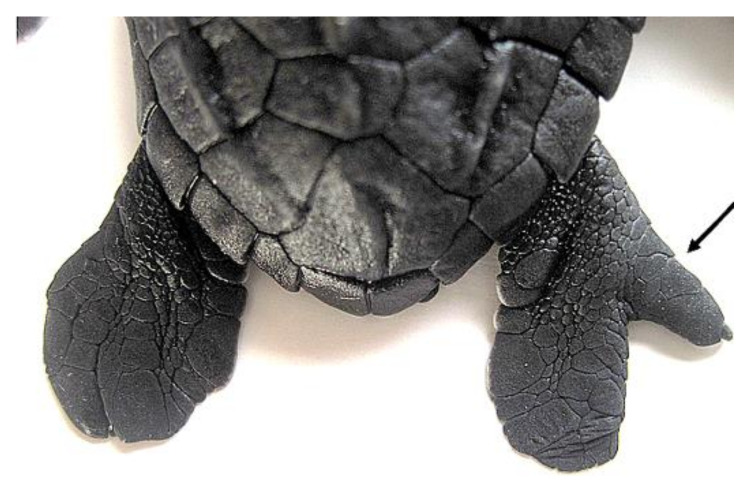
*Lepidochelys olivacea* showing syndactyly (arrow, dorsal view). Photograph taken from supplementary file in [13] with the permission of the authors and publisher (John Wiley & Sons).

**Table 1 animals-11-00444-t001:** Summary of embryonic development of sea turtles (stages, processes, and main morphological traits) according to [10,11,12] and prevalent affectations in sea turtles due to teratogenic agents. Images from Miller et al. [12] with the permission of the author and the publisher (CCB, Allen Press Publishing Services); scale bar = 1 mm for stages 7–20; 5 mm for stage 25; 10 mm for stages 26–31; arrows indicate key characteristics.

Developmental Stage	Developmental Process	Morphological Traits	Prevalent Affectations in Sea Turtles Due to Teratogenic Agents
1–9 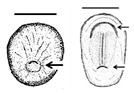	*Fertilisation-Gastrulation-Early Neurulation* After fertilisation, the zygote undergoes cleavage to form a blastocyst; the blastula is reorganised into a trilaminar structure. Neurulation initiates	Primitive streak; trilaminar embryonic disc composed of endoderm, ectoderm, and mesoderm; notochordal plate; neural groove; neural folds; head fold	The egg is rich in lipid proteins and is ovoposited in middle gastrula stage; depending on the exposure to a teratogenic factors (such as extreme temperatures), embryonic development may be halted [10,11,12]
10–14 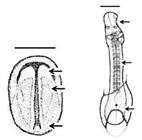	*Neurulation-Somitogenesis* The neural plate folds and transforms into the neural tube; somites form from the paraxial mesoderm of the neurulating embryo	Neural tube; somites (up to 17); amnion covers half of the embryo; germ layers; amniotic cap; otic vesicle; first pharyngeal cleft; development of the head, brain, heart, and blood vessels	Gnathoschisis (cleft palate) is the prevalent craniofacial malformation in green, loggerhead and olive ridley sea turtles [13,19] caused by an incorrect fusion of the medial nasal and maxillary prominences and frontonsal mass deriving from mesenchymal and neural crest cells [26]
15–19 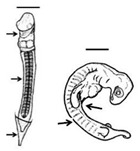	*Somitogenesis-Organogenesis* Organised and integrated processes by which embryonic layers transform into tissues and organs	Torsion initiates and completes; embryo on left side; somites (>40); amnion, chorion and yolk sac complete; pharyngeal clefts open and start to close; limb buds develop to form digital plates; blood islands visible; tail elongates; lens visible in the eye	Dysmelia (limb differences) is the most common malformation reported for hawksbill turtles [13]; deviations in the signal centre of the apical ectodermal ridge (AER) cause truncated limb bones [27,28]
20–25 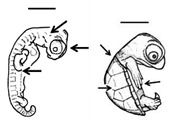	*Early growth* All major organs and systems continue to develop and grow	All pharyngeal clefts closed; iris pigmented along posterior border; limbs develop; ribs are visible; carapace develops with scutes; tail equals hindlimbs in length, organs and systems continue to grow	Variation in scute patterns is the most common malformation in hard-shelled sea turtles, and is compatible with life [10,11,13]; hot temperatures and dry conditions may produce scute variations [29,30]
26–31 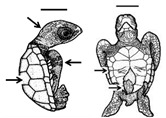	*Late growth-Hatching* Organs and systems conclude growth and development; a fully developed sea turtle is ready to hatch	Head and flipper scales present; diameter of yolk decreases (piping stage <10 mm); all organs and systems fully developed; pigmentation is evident; hatching occurs	Leucism (hypopigmentation) has been reported for several species of turtles [13,21,22,23,24,25]; these individuals can survive to adulthood, and remain reproductively active [31]

## Data Availability

No new data were created or analyzed in this study. Data sharing is not applicable to this article.

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
