# Peer review of "Congenital Malformations in Sea Turtles: Puzzling Interplay between Genes and Environment"

_animals, 2021, doi:10.3390/ani11020444_

Round 1

Reviewer 1 Report

This is one of the most comprehensive and insightful reviews I have read as of late.  Many researchers have discussed congenital anomalies in turtles; however, the work presented here provides a synthesis to address aberrations in the entire organism.  It is interesting to speculate and hypothesize that mechanisms known from other organisms may play a role in similar anomalies in turtles, something which may be studied more extensively thanks to genome and transcriptome sequencing efforts.  Integrating evidence from DNA methylation and environmental effects also makes the hypotheses proposed timely.

I find the figures to be well thought out and useful for readers even beyond the development of turtles.  Also extending beyond turtles to evidence from other reptiles and discussing monitoring programmes are useful integrations.

I have very minor comments and look forward to seeing this work published.  If the journal highlights articles, I recommend this one, as it will be of very broad interest.

  1. "Insults" - this does not sit well. Factor? Source?
  2. lines 202-203: the authors note that sustained high temperatures may cause neural tube defects in these species - I would further add information regarding studies where higher temperatures during sea turtle development have been documented but did not result in neural tube defects (Zimm et al. 2017 doi:10.1093/icb/icx066 your reference 104). Though not necessary to cite, perhaps the authors would find Bentley et al. 2017 (doi: 10.1111/mec.14087) useful as an attempt to understand developmental processes that may be compromised at higher temperatures.
  3. line 455: replace "It develops from primary signalling centres along the carapacial ridge" with "Turtle shell development begins with the formation of a specialized structure, the carapacial ridge," and I would cite Burke 1989 (doi.org/10.1002/jmor.1051990310), Moustakas-Verho et al. 2017 (dx.doi.org/10.1016/j.gde.2017.03.016).  After you describe the ribs, before line 460 Thus, you may add that the shell is covered by an array of modular epidermal structures called scutes that develop from primary signalling centres along the carapacial ridge (Moustakas-Verho et al. 2014 - your reference 101).  Line 463 "located at THE carapacial ridge" (missing t in the).
  4. Line 545: a recent publication by Bentley et al. 2020 (DOI:10.1002/jmor.21294) shows that there is no evidence that these scute variations affect survival.

Author Response

The authors would like to thank the reviewers for taking their time to review this document and make constructive comments and suggestions to this manuscript. There were substantial changes to the document; all of them are highlighted in yellow, as well as the additional references.

Reviewer 1

This is one of the most comprehensive and insightful reviews I have read as of late.  Many researchers have discussed congenital anomalies in turtles; however, the work presented here provides a synthesis to address aberrations in the entire organism.  It is interesting to speculate and hypothesize that mechanisms known from other organisms may play a role in similar anomalies in turtles, something which may be studied more extensively thanks to genome and transcriptome sequencing efforts.  Integrating evidence from DNA methylation and environmental effects also makes the hypotheses proposed timely.

Thank you.

I find the figures to be well thought out and useful for readers even beyond the development of turtles.  Also extending beyond turtles to evidence from other reptiles and discussing monitoring programmes are useful integrations.

I have very minor comments and look forward to seeing this work published.  If the journal highlights articles, I recommend this one, as it will be of very broad interest.

  1. "Insults" - this does not sit well. Factor? Source?

We changed “insults” for “factors”, “alterations”, or “deviations” depending on the context.

  1. lines 202-203: the authors note that sustained high temperatures may cause neural tube defects in these species - I would further add information regarding studies where higher temperatures during sea turtle development have been documented but did not result in neural tube defects (Zimm et al. 2017 doi:10.1093/icb/icx066 your reference 104). Though not necessary to cite, perhaps the authors would find Bentley et al. 2017 (doi: 10.1111/mec.14087) useful as an attempt to understand developmental processes that may be compromised at higher temperatures.
  2. line 455: replace "It develops from primary signalling centres along the carapacial ridge" with "Turtle shell development begins with the formation of a specialized structure, the carapacial ridge," and I would cite Burke 1989 (doi.org/10.1002/jmor.1051990310), Moustakas-Verho et al. 2017 (dx.doi.org/10.1016/j.gde.2017.03.016).  After you describe the ribs, before line 460 Thus, you may add that the shell is covered by an array of modular epidermal structures called scutes that develop from primary signalling centres along the carapacial ridge (Moustakas-Verho et al. 2014 - your reference 101).  Line 463 "located at THE carapacial ridge" (missing t in the).
  3. Line 545: a recent publication by Bentley et al. 2020 (DOI:10.1002/jmor.21294) shows that there is no evidence that these scute variations affect survival.

For points 2, 3, and 4, we have revised all the articles suggested by the reviewer, and modified and added the information and references accordingly, thank you very much for your suggestions.

Reviewer 2 Report

See the attached MS for additional editing and comments

Editing by the authors is poor. The summary and abstract contain examples of the awkward use of words and sentences that detract from the communication of the ideas.  The language expression is poor.  I do not make these comments to be overly picky.  Rather, I want you to communicate your ideas well because the understanding of malformations is an important, but under studied topic.

In lines 30/31:  if congenital malformations are ‘rarely observed’ then Line 35/36, you say “among wild animals” (as opposed to tame ones?); I think you mean to say something like: malformations are rarely observed in free-ranging populations.  You use the phrase “presenting mortality . . . ” (The language is stilted).  Consider: “Oviparous reptiles, such as sea turtles, that produce numerous eggs in a clutch that is buried in the beach provide an opportunity to examine embryonic mortality associated with malformations that occur at different times during development or that prevent the hatchling from emerging from the nest.”

Line 39: . . . was performed (conducted) along the {place / area} coast of Mexico . .

Line 40: Why is it important to state “some unpublished . . “  at this point in the text?

Line 40:  Provide the percentage of ‘craniofacial malformations.

Line 41: Provide a percentage.  You must also use parallel construction in the sentence structure:  ‘. . several malformations in volving the limbs, carapace and pigmentation were observed, indicating mortality occurred during the middle and later stages of development’.

Line 43-45: detract from the flow of your text.

Line 56: . . . ‘is the study of the causes . . ‘  OR ‘ . . is the scientific study of the . . ‘

Line 58: ‘The causes of developmental malformations . . .’

Line 62-64:  Consider: Teratogens may . . . malformations at any time during development.  Depending on the severity of the impact of teratogens on the developmental process, the embryo may die or be malformed.  Depending on the severity of the malformation the embryo may hatch, and the hatchling may emerge from the nest.  Severely malformed . . .’

Line 73: ‘. . until hatchling emergence . . ‘   But until this point you have been stressing the impact on development of teratogenic factors and now you say development depends on environmental factors ‘ [you need to express the ideas being presented more fully. 

Line 77:  Australian

Lines 75-80:  I do not understand the positioning of this paragraph.  It disrupts the presentation of the information by jumping to a species

Line 81-85:  . . . reported finding several types of malformations . . [phrasing is awkward and poorly placed which changes the emphasis of the ideas in the sentence]

Line 84:  are you suggesting that the % malformation depended on the species rather than teratogenic factors? The percentage also depends on the numbers of embryos being examined.

Line 73 & 105- 109 [Table 1]:  I do not see the point of including Table 1 because it is available on-line and is not the primary source of the descriptions of embryonic development in sea turtles [reference # 7 is the primary source of descriptions].  It is only cited once in the text (line 73).  Its inclusion does not advance the goals of the paper as stated in lines 102-104.

Line 102; expression: why use the past tense?  identify and describe.   You do provide descriptions and comparison to mammalian teratology, some of which reads very well

Line 137: change in citation style; compare to style used in line 81

Line 141: than

Line 142: unsupported speculation

Line 145:  ‘in’ should be  “concerning . . . . sea turtle embryos, craniofacial . . “. 

Line 165: evidenced [noun being used as a verb].  Change to shown or illustrated

Line 168: see comment about style of citation in text.   see Line 178 for proper style [rather than just citing a number when the authors are part of the sentence.

Line 174: but may hatch.

Line 174: you offer no proof of the statement concerning the exposure of embryos to ‘environmental factors nor concerning the impact of these factors on development.

Line 185: i.e., Zika

Line 187:  Because [not Since]

General comment: pictures could be placed side by side to conserve journal page space.

Line 197:  . . .(meninges).  It is caused

Line 222:  A leatherback sea turtle embryo with . . .  [the phrasing used is awkward]

Line 242 – 243: if you remove the phrase ‘, as well as other pathologies, ‘  the sentence is missing some words.  This is a problem associated with using subordinate clauses (i.e., the writer and reader lose track of what is being said).

Line 249:  break the two sentences apart at the semicolon ( ; )

Line 269: replace coma with full stop

Line 270: separate sentence at the semicolon.

Line 291:  in olive ridley turtle embryos

line 295: in olive turtle embryos

line 302: embryos

line 315: either use a semicolon of separate into two sentences

line 340:  compared to malformations of the

Line 467: separate into two sentences

Line 471 citation needed

Line 472 citation needed

Lines 473-479: run-on sentence that should be split into at least two in not three sentences

Line 480; Split into two sentences and provide citations

Line 485:  the word embryo should be repeated to remind the reader that what is being said refers to embryos and not other life stages.

Line 494: Split into two sentences

Line 498: Split into two sentences

Line 500: Split into two sentences

Line 508 citation required

Line 511:  citation required

Line 516 words missing

Line 512-517:  this single sentence paragraph should be broken into several sentences and a better justification provided as to why the focus is limited whereas comments above have not been limited.

Line 518:  do you mean mammals? Other reptiles?  Citation required

Line 519:  split into two sentences and include the idea of the stage of development of the embryo (i.e., whether the embryological processes occurring at that time are sensitive or not).  The idea is picked up a few sentences along.  This paragraph should be rewritten to better organize the flow of the ideas.

Line 524: do you mean duration and actual temperature involved?

Line 541: produced

Line 547: citation needed

Line 552 citation needed

Line 553-554:  reviewed, including their effects of

Line 577-578:  you have jumped to non-embryonic life stages and have provided no explanatory comment.  Either develop the idea more or deleted it

Line 587 split into two sentences

Line 598: these citations belong at the end of the sentence

Line 605: in a decrease in the maximum daily temperature    preferred . . . was . .

 [in addition, this study involved turtles in captivity (not iin the wild)

Line 607: verb tense problem

Line 608 in subsequent dead embryos?  Live hatchlings?  You need to explain the context and the results more carefully.

Line 610-612: your shaded recommendation is not justified by the information you have presented.  ‘Due to’ should be ‘resulting from’.  You have not defined what is meant by ‘heat shock’

Lines 613-618: you say temperature has been studied more that humidity

then lines 620 -627 you talk about temperature and do not discuss the measurement or provide examples of the impact of humidity.

Then beginning on line 638 you discuss the quality of foraging areas.  The paragraph does not contain a citation yet you have jumbled multiple ideas into it. 

Summary:  This is an ambitious review that says it is going to review some very important information.

The abstract suggests that the information to be presented comes from a “extensive field survey” (lines 37-37) yet no methods are presented, no numerical results from the examination of multiple remnants of nests exhumed are presented.  The data were apparently presented in citation [9] (line 81). Based on lines 102-104, I expected a recapitulation of malformations and their frequency along with a discussion of possible causes. Although there is a presentation of possible causes, no data are presented to support the inclusion of the “most common malformations”. 

I have marked several grammatical issues and expression issues but not all. These need to be corrected.

The text and the explanations need considerably more work.  If my comments read as being harsh, please understand that you have the basis of a very good review and I would like to see you revise it.

Author Response

The authors would like to thank the reviewers for taking their time to review this document and make constructive comments and suggestions to this manuscript. There were substantial changes to the document; all of them are highlighted in yellow, as well as the additional references.

Reviewer 2

See the attached MS for additional editing and comments.

It was very useful, thank you.

Editing by the authors is poor. The summary and abstract contain examples of the awkward use of words and sentences that detract from the communication of the ideas.  The language expression is poor.  I do not make these comments to be overly picky.  Rather, I want you to communicate your ideas well because the understanding of malformations is an important, but under studied topic.

The document was and revised by a native English speaker. The Summary and Abstract were edited, we tried to use the language correctly (without awkward use of words and sentences), and tried to improve the language expression, and we hope we did better this time.

In lines 30/31:  if congenital malformations are ‘rarely observed’ then Line 35/36, you say “among wild animals” (as opposed to tame ones?); I think you mean to say something like: malformations are rarely observed in free-ranging populations.  You use the phrase “presenting mortality . . . ” (The language is stilted).  Consider: “Oviparous reptiles, such as sea turtles, that produce numerous eggs in a clutch that is buried in the beach provide an opportunity to examine embryonic mortality associated with malformations that occur at different times during development or that prevent the hatchling from emerging from the nest.”

We followed your suggestions; these lines in the Abstract were modified accordingly.

Line 39: . . . was performed (conducted) along the {place / area} coast of Mexico . .

This information has been added/corrected in the Abstract.

Line 40: Why is it important to state “some unpublished . . “  at this point in the text?

This statement was eliminated.

Line 40:  Provide the percentage of ‘craniofacial malformations.

The percentage of craniofacial malformations (as well as other malformations in the body) was provided.

Line 41: Provide a percentage.  You must also use parallel construction in the sentence structure:  ‘. . several malformations involving the limbs, carapace and pigmentation were observed, indicating mortality occurred during the middle and later stages of development’.

This information was added/corrected in the Abstract and Introduction sections.

Line 43-45: detract from the flow of your text.

The sentence was deleted.

Line 56: . . . ‘is the study of the causes . . ‘  OR ‘ . . is the scientific study of the . . ‘

“…is the study of the causes…” it has been corrected.

Line 58: ‘The causes of developmental malformations . . .’

The sentence was modified according to this suggestion.

Line 62-64:  Consider: Teratogens may . . . malformations at any time during development.  Depending on the severity of the impact of teratogens on the developmental process, the embryo may die or be malformed.  Depending on the severity of the malformation the embryo may hatch, and the hatchling may emerge from the nest.  Severely malformed . . .’

The text was modified accordingly.

Line 73: ‘. . until hatchling emergence . . ‘   But until this point you have been stressing the impact on development of teratogenic factors and now you say development depends on environmental factors ‘ [you need to express the ideas being presented more fully. 

The text was modified accordingly.

Line 77:  Australian

Australian was corrected.

Lines 75-80:  I do not understand the positioning of this paragraph.  It disrupts the presentation of the information by jumping to a species

The entire paragraph was removed and restructured.

Line 81-85:  . . . reported finding several types of malformations . . [phrasing is awkward and poorly placed which changes the emphasis of the ideas in the sentence]

The text was edited.

Line 84:  are you suggesting that the % malformation depended on the species rather than teratogenic factors? The percentage also depends on the numbers of embryos being examined.

Apparently some species of sea turtles are more prone to congenital malformations; nevertheless we eliminated “depending on the species” and added the number of embryos that we examined.

Line 73 & 105- 109 [Table 1]:  I do not see the point of including Table 1 because it is available on-line and is not the primary source of the descriptions of embryonic development in sea turtles [reference # 7 is the primary source of descriptions].  It is only cited once in the text (line 73).  Its inclusion does not advance the goals of the paper as stated in lines 102-104.

We decided to leave Table 1 because it describes normal development, which could be important to include, however, we included “Prevalent affectations in sea turtles due to teratogenic agents” and it is cited several times in the text.

Line 102; expression: why use the past tense?  identify and describe.   You do provide descriptions and comparison to mammalian teratology, some of which reads very well

The reviewer is right. We modified the text as “…we describe…”

Line 137: change in citation style; compare to style used in line 81

The citation style was corrected.

Line 141: than

“that” was changed for “than”.

Line 142: unsupported speculation

This was not our speculation; we modified the text, relocated the references, and included a new reference.

Line 145:  ‘in’ should be  “concerning . . . . sea turtle embryos, craniofacial . . “. 

“in” was changed for “concerning”.

Line 165: evidenced [noun being used as a verb].  Change to shown or illustrated

“evidenced” was changed for “illustrated”.

Line 168: see comment about style of citation in text.   see Line 178 for proper style [rather than just citing a number when the authors are part of the sentence.

The citation style was corrected.

Line 174: but may hatch.

The text was edited.

Line 174: you offer no proof of the statement concerning the exposure of embryos to ‘environmental factors nor concerning the impact of these factors on development.

The text was edited and we have provided some information as an example: “…bicephaly is by far the most reported malformation in snakes, both wild and in captivity, and associated causes could be both cool and hot temperatures during incubation, inbreeding depression, hybridization, environmental pollution, and chemical toxins”

Line 185: i.e., Zika

Zika was corrected.

Line 187:  Because [not Since]

“Since” was changed for “Because”

General comment: pictures could be placed side by side to conserve journal page space.

The reviewer is right. We placed pictures side by side.

Line 197:  . . .(meninges).  It is caused

We corrected the sentence as suggested.

Line 222:  A leatherback sea turtle embryo with . . .  [the phrasing used is awkward]

We corrected the sentence as suggested.

Line 242 – 243: if you remove the phrase ‘, as well as other pathologies, ‘  the sentence is missing some words.  This is a problem associated with using subordinate clauses (i.e., the writer and reader lose track of what is being said).

The paragraph was edited: “Interestingly, lower blood cholesterol levels have been reported for green turtles with fibropapilloma (as well as with other pathologies) compared to healthy turtles [57-59]. Blood cholesterol has been recently proposed as a biomarker of fibropapillomatosis in green turtles…”

Line 249:  break the two sentences apart at the semicolon ( ; )

We changed semicolon for full stop.

Line 269: replace coma with full stop

We replaced comma with full stop.

Line 270: separate sentence at the semicolon.

We separated the sentence at the semicolon.

Line 291:  in olive ridley turtle embryos

line 295: in olive turtle embryos

line 302: embryos

We added the word “embryos” whenever was necessary in the text, as suggested.

line 315: either use a semicolon of separate into two sentences

A semicolon was used.

line 340:  compared to malformations of the

The sentence was edited as suggested.

Line 467: separate into two sentences

The text was edited and separated in two sentences.

Line 471 citation needed

Line 472 citation needed

The text was edited.

Lines 473-479: run-on sentence that should be split into at least two in not three sentences

The sentence was split in two sentences.

Line 480; Split into two sentences and provide citations

The sentence was split in two and the reference was provided.

Line 485:  the word embryo should be repeated to remind the reader that what is being said refers to embryos and not other life stages.

We repeated the word “embryo” as suggested.

Line 494: Split into two sentences

Line 498: Split into two sentences

Line 500: Split into two sentences

All sentences were split.

Line 508 citation required

Line 511:  citation required

Citations were included.

Line 516 words missing

The text was edited.

Line 512-517:  this single sentence paragraph should be broken into several sentences and a better justification provided as to why the focus is limited whereas comments above have not been limited.

The paragraph was edited as suggested. “…we focus this discussion on temperature, humidity, and anthropogenic-derived contaminants (i.e. persistent organic compounds such as pesticides and hydrocarbons, widely distributed in the aquatic environment), because these factors have been monitored in both feeding and nesting areas, providing useful information to help improve management.”

Line 518:  do you mean mammals? Other reptiles?  Citation required

Line 519:  split into two sentences and include the idea of the stage of development of the embryo (i.e., whether the embryological processes occurring at that time are sensitive or not).  The idea is picked up a few sentences along.  This paragraph should be rewritten to better organize the flow of the ideas.

Line 524: do you mean duration and actual temperature involved?

For these three points, The paragraph was edited and citations were included as required: “Hyperthermia has been regarded as teratogenic in different mammalian species [120-122]. The teratogenic effect of extreme temperatures depends on timing, intensity, duration, and actual temperature involved [120]. The incidence of congenital malformations depends on the species and the embryological stage at the time of exposure [121,122]. In most mammalian species, hyperthermia is reached between 1.5 and 2.5°C above the normal body temperature; in fact, 1.5°C above normal temperature in the pre-implantation embryo may result in a high mortality rate, whereas after implantation may cause congenital malformations [123]. Temperature-associated malformations include neural tube defects, microphtalmia, microcephaly, limb defects, craniofacial malformations, and developmental defects of internal organs such as abdominal wall, heart, and kidneys [120].“

Line 541: produced

The sentence was edited.

Line 547: citation needed

Line 552 citation needed

Citations were indicated.

Line 553-554:  reviewed, including their effects of

The sentence was corrected as suggested.

Line 577-578:  you have jumped to non-embryonic life stages and have provided no explanatory comment.  Either develop the idea more or deleted it

The reviewer is right. The sentence was deleted because it was not related to embryonic life stages.

Line 587 split into two sentences

We changed the comma for a semicolon because it was part of the same idea…

Line 598: these citations belong at the end of the sentence

The citations were relocated at the end of the sentence.

Line 605: in a decrease in the maximum daily temperature    preferred . . . was . .

 [in addition, this study involved turtles in captivity (not iin the wild)

The sentence was edited.

Line 607: verb tense problem

Line 608 in subsequent dead embryos?  Live hatchlings?  You need to explain the context and the results more carefully.

The paragraph was edited.

Line 610-612: your shaded recommendation is not justified by the information you have presented.  ‘Due to’ should be ‘resulting from’.  You have not defined what is meant by ‘heat shock’

The sentence was edited: “…Shade cover is highly recommended to lower hot temperatures during the summer, reducing the effects of hyperthermia during critical stages of development in sea turtles.”

Lines 613-618: you say temperature has been studied more that humidity then lines 620 -627 you talk about temperature and do not discuss the measurement or provide examples of the impact of humidity.

The sentence was edited: “…whereas humidity is managed by a sand sanitisation procedure involving sand screening, aeration, flattening, and water sprinkling…”

Then beginning on line 638 you discuss the quality of foraging areas.  The paragraph does not contain a citation yet you have jumbled multiple ideas into it. 

The reviewer is right. Citations were provided.

 Summary:  This is an ambitious review that says it is going to review some very important information.

The abstract suggests that the information to be presented comes from a “extensive field survey” (lines 37-37) yet no methods are presented, no numerical results from the examination of multiple remnants of nests exhumed are presented.  The data were apparently presented in citation [9] (line 81). Based on lines 102-104, I expected a recapitulation of malformations and their frequency along with a discussion of possible causes. Although there is a presentation of possible causes, no data are presented to support the inclusion of the “most common malformations”. 

Thank you for all your suggestions. The Abstract was modified and information was added to Table 1 and throughout the text.

I have marked several grammatical issues and expression issues but not all. These need to be corrected.

The text and the explanations need considerably more work.  If my comments read as being harsh, please understand that you have the basis of a very good review and I would like to see you revise it.

We truly appreciate the time you took to review this manuscript with such detail. We hope we understood every comment and improved the contents of this document. Thank you.

Round 2

Reviewer 1 Report

The revision is appreciated, thank you for this interesting paper.

Author Response

The revision is appreciated, thank you for this interesting paper.

We thank you for your constructive comments.

Reviewer 2 Report

Because it presents a catalog of malformations found in marine turtle embryos and hatchlings that have impacted development and relates those malformations to potential and known causes, I believe the paper brings together interesting and useful information.  Although the bulk of the manuscript has been revised according to most of the suggestions made by the reviewers, I believe that the paper still needs some revision. Below are some examples of issues that need to be addressed.

Line 60: Sperm are stored in the ‘middle of the oviduct’.  Because sperm storage does not occur in the same place in all groups of reptiles, reference should be made to papers that describe where sperm are stored in turtles (for review see: Girling 2002).  Fertilization must occur in the infundibulum (or in the anterior uterine tube) before albumen deposition begins.

Girling, J. E. (2002). The reptilian oviduct: a review of structure and function and directions for future research. J Exp Zool 293(2): 141-170.

 Lines 622 – 624 add a new idea to the paper about the impact of shading on the behaviour of adult females in captivity; yet lines 615 – 621 and lines 624 – 626 focus on temperature impacts during development. Lines 622 – 624 cause a disruption in the flow of logic of the text and, unless reference 157 makes statements about shade and hyperthermia in embryos in nests, they should be deleted. If reference 157 does, then more information must be provided that links between the temperature of adult females and hyperthermia in developing embryos.

line 661 “strict regulations should be implemented to reduce . . ” ; delete “in order”.  Can you provide a reference that demonstrates “increasing availability of high quality food . . .”?  You need to offer some credible information that supports what you are asserting.

Similarly, line 664, Can you provide a reference that demonstrates that reducing human activity near nesting beaches actually improves offspring survival and fitness?

 Line 663 – 664 Why say “inbreeding areas” followed by “(in proximity of nesting beaches)”? delete one or the other.

There are still word usage issues that should be addressed. For example, in line 502 the word ‘since’ should be ‘because’ and ‘emerge’ should be ‘be visible’.  Word choice is critical to communication, especially so when dealing with embryonic processes.

The objectives of the paper are stated in lines 42 – 44. The first part of the objectives is achieved by the descriptions and the discussions presented in the body of the text. However, the presentation of recommendations to reduce embryonic mortality are not well linked to management action described in the published literature. Nor are the actions and their results described in detail in the body of the text. The one example presented is from Mexico on how to manage for improved hatching success and a low rate of embryonic malformation.  However, nothing in what is presented in lines 634-642 demonstrates that the occurrence and rate of malformations resulted in a reduction in malformations.  Table #2 lists hatching success but not a rate of malformation. I suggest that you put all the details about this project into another short paper or put the information and pictures into the Supplementary information attachment.

I reiterate that my comments are meant to help you improve the paper.  I encourage you to re-read the manuscript very carefully with the intention of improving it.  I look forward to it in print.

Author Response

We thank the reviewer for all the constructive comments and suggestions to this document. All modifications to the text are highlighted in yellow. We have answered point by point:

 Because it presents a catalog of malformations found in marine turtle embryos and hatchlings that have impacted development and relates those malformations to potential and known causes, I believe the paper brings together interesting and useful information.  Although the bulk of the manuscript has been revised according to most of the suggestions made by the reviewers, I believe that the paper still needs some revision. Below are some examples of issues that need to be addressed.

Line 60: Sperm are stored in the ‘middle of the oviduct’.  Because sperm storage does not occur in the same place in all groups of reptiles, reference should be made to papers that describe where sperm are stored in turtles (for review see: Girling 2002).  Fertilization must occur in the infundibulum (or in the anterior uterine tube) before albumen deposition begins.

Girling, J. E. (2002). The reptilian oviduct: a review of structure and function and directions for future research. J Exp Zool 293(2): 141-170.

 The text was edited: “…females can store viable sperm to fertilise more eggs; fertilization occurs in the upper oviduct and although no specialized sperm storage structures are present in sea turtles, sperm is stored in the ducts of albumin glands in the upper region of the oviduct…” Girling, 2002 and other references were included. 

 Lines 622 – 624 add a new idea to the paper about the impact of shading on the behaviour of adult females in captivity; yet lines 615 – 621 and lines 624 – 626 focus on temperature impacts during development. Lines 622 – 624 cause a disruption in the flow of logic of the text and, unless reference 157 makes statements about shade and hyperthermia in embryos in nests, they should be deleted. If reference 157 does, then more information must be provided that links between the temperature of adult females and hyperthermia in developing embryos.

 Reference 152 was revised and the authors do not relate temperature in adult females with hyperthermia in embryos, the study lasted 2 months and was performed in juveniles between 26 and 29 cm CCW. For this reason, the statements (and the reference) were eliminated from the review. 

line 661 “strict regulations should be implemented to reduce . . ” ; delete “in order”.  Can you provide a reference that demonstrates “increasing availability of high quality food . . .”?  You need to offer some credible information that supports what you are asserting.

 The paragraph was edited and references were added.

Similarly, line 664, Can you provide a reference that demonstrates that reducing human activity near nesting beaches actually improves offspring survival and fitness?

 We do not have a reference demonstrating that reducing human activity near nesting beaches actually improves offspring survival and fitness, so we were more careful with the statement: “Based on results from monitoring programmes, regulations should be implemented to reduce human activities in feeding and breeding areas, to lower the presence of possible anthropogenic teratogens in marine ecosystems that could eventually be transferred to the embryos, and to avoid the formation of endogamic populations that may express abnormal phenotypes.”

 Line 663 – 664 Why say “inbreeding areas” followed by “(in proximity of nesting beaches)”? delete one or the other.

 The sentence was edited as suggested.

 There are still word usage issues that should be addressed. For example, in line 502 the word ‘since’ should be ‘because’ and ‘emerge’ should be ‘be visible’.  Word choice is critical to communication, especially so when dealing with embryonic processes.

The words “since” and “emerge” were changed for “because” and “be visible” as suggested.

The objectives of the paper are stated in lines 42 – 44. The first part of the objectives is achieved by the descriptions and the discussions presented in the body of the text. However, the presentation of recommendations to reduce embryonic mortality are not well linked to management action described in the published literature. Nor are the actions and their results described in detail in the body of the text. The one example presented is from Mexico on how to manage for improved hatching success and a low rate of embryonic malformation.  However, nothing in what is presented in lines 634-642 demonstrates that the occurrence and rate of malformations resulted in a reduction in malformations.  Table #2 lists hatching success but not a rate of malformation. I suggest that you put all the details about this project into another short paper or put the information and pictures into the Supplementary information attachment.

 The reviewer is right; we cannot demonstrate that the management procedure in the beach resulted in a reduction in malformations. We have moved this information to the Supplementary file. Nevertheless we included in Table S1 the number of relocated nests, number of hatchlings, and number of malformed embryos.

I reiterate that my comments are meant to help you improve the paper.  I encourage you to re-read the manuscript very carefully with the intention of improving it.  I look forward to it in print.

 Thank you for your comments; this is the purpose of peer-review, constructive criticism, exactly like this.